# The SARS-CoV-2 S1 Spike Protein Promotes MAPK and NF-kB Activation in Human Lung Cells and Inflammatory Cytokine Production in Human Lung and Intestinal Epithelial Cells

**DOI:** 10.3390/microorganisms10101996

**Published:** 2022-10-10

**Authors:** Christopher B. Forsyth, Lijuan Zhang, Abhinav Bhushan, Barbara Swanson, Li Zhang, João I. Mamede, Robin M. Voigt, Maliha Shaikh, Phillip A. Engen, Ali Keshavarzian

**Affiliations:** 1Department of Internal Medicine, Section of Gastroenterology, Rush Center for Integrated Microbiome and Chronobiology Research, Department of Anatomy and Cell Biology, Rush University Graduate College, Rush University Medical Center, Chicago, IL 60612, USA; 2Rush Center for Integrated Microbiome and Chronobiology Research, Rush University Medical Center, Chicago, IL 60612, USA; 3Department of Biomedical Engineering, Illinois Institute of Technology, Chicago, IL 60616, USA; 4Department of Adult Health & Gerontological Nursing, Rush University Medical Center, Chicago, IL 60612, USA; 5Department of Microbial Pathogens and Immunity, Rush University Medical Center, Chicago, IL 60612, USA

**Keywords:** COVID-19, ARDS, cytokine storm, S1 spike protein, SARS-CoV-2, PASC, ACE2, A549 cells, Caco-2 cells

## Abstract

The coronavirus disease 2019 (COVID-19) pandemic began in January 2020 in Wuhan, China, with a new coronavirus designated SARS-CoV-2. The principal cause of death from COVID-19 disease quickly emerged as acute respiratory distress syndrome (ARDS). A key ARDS pathogenic mechanism is the “Cytokine Storm”, which is a dramatic increase in inflammatory cytokines in the blood. In the last two years of the pandemic, a new pathology has emerged in some COVID-19 survivors, in which a variety of long-term symptoms occur, a condition called post-acute sequelae of COVID-19 (PASC) or “Long COVID”. Therefore, there is an urgent need to better understand the mechanisms of the virus. The spike protein on the surface of the virus is composed of joined S1–S2 subunits. Upon S1 binding to the ACE2 receptor on human cells, the S1 subunit is cleaved and the S2 subunit mediates the entry of the virus. The S1 protein is then released into the blood, which might be one of the pivotal triggers for the initiation and/or perpetuation of the cytokine storm. In this study, we tested the hypothesis that the S1 spike protein is sufficient to activate inflammatory signaling and cytokine production, independent of the virus. Our data support a possible role for the S1 spike protein in the activation of inflammatory signaling and cytokine production in human lung and intestinal epithelial cells in culture. These data support a potential role for the SARS-CoV-2 S1 spike protein in COVID-19 pathogenesis and PASC.

## 1. Introduction

From December 2019 to January 2020, a pneumonia illness originating in Wuhan, China, designated as coronavirus disease 2019 (COVID-19) was shown to be caused by a novel RNA coronavirus, designated as severe acute respiratory syndrome coronavirus 2 (SARS-CoV-2) [1,2,3]. The World Health Organization (WHO) declared COVID-19 an international health emergency in late January 2020 [4,5]. It is estimated that the current COVID-19 pandemic has affected over 603 million people, with more than 6.4 million deaths worldwide. As of August 2022, the United States has accounted for over 95 million cases and over 1 million deaths [6]. Coronaviruses, including SARS-CoV-1 (SARS), MERS-CoV, and SARS-CoV-2, are respiratory viruses that have acute respiratory distress syndrome (ARDS) as the major cause of death. ARDS is a severe and often fatal form of acute lung injury that occurs in critically ill patients [7,8]. Severe COVID-19 disease is associated with a so-called “Cytokine Storm” [9,10,11], which is a dramatic increase in inflammatory cytokines/chemokines (especially IL-1β, IL-6, IL-8, and TNF-α) and other inflammatory factors that promote tissue injury and can predict the severity of COVID-19 and survival [9].

A large body of data now supports diverse COVID-19 pathology. COVID-19 not only causes respiratory disease; it may also cause gastrointestinal, cardiovascular, and neurological disease [12,13,14]. These diverse pathologies are apparent in acute COVID-19 disease, as well as in a condition known as post-acute sequelae of COVID-19 (*PASC)*, which is also known as “Long COVID” or “long haulers”, a condition that persists beyond six months even in the absence of a positive test for the virus [14,15,16]. One emerging candidate that possibly contributes to the acute effects of COVID-19 and PASC is the S1 region of the SARS-CoV-2 spike protein [17,18,19]. In SARS-CoV-2 infection, the “spike” (S) protein found on the surface of the coronavirus binds to the angiotensin-converting enzyme 2 (ACE2) receptor on human cells to gain entry by a cleavage/endocytosis-dependent process [20,21,22]. The spike proteins on the virus surface give the virus the “corona” appearance that is characteristic of coronaviruses, and the S protein is the target of some vaccines. Each surface spike receptor is a trimer, composed of three identical spike protein monomers that are tightly associated with each other, and each monomer contains two distinct joined subunits: S1 and S2 [23,24]. The S1 outer subunit contains the “receptor-binding-domain” (RBD) region that provides the binding site and specificity for ACE2 [25,26]. After S1 subunit/receptor binding to ACE2, the region between S1 and S2 is cleaved to release S1, and S2 mediates viral fusion to the cell membrane and viral entry [22]. Thus, the S1 spike subunit is released with each cellular infection and the S1 protein is also shed spontaneously [27]. One study estimates 300:1-fold greater numbers of free S1 spike in the blood during infection vs. virus in COVID-19 disease [28]. While the spike protein in COVID-19 was previously assumed not to have independent biological activity [24], recent data show that high S1 spike protein levels correlate with poor prognosis of COVID-19, ICU admission, and death [29,30]. Thus, the S1 spike protein might be one pivotal trigger for the initiation and/or perpetuation of the cytokine storm and PASC.

The present study assessed the impact of the S1 protein on human lung and intestinal epithelial cells as important organ systems impacted in COVID-19 and PASC. We sought to test the hypothesis that the S1 spike protein is sufficient to promote inflammation, including activation of MAPK (ERK1/2) and/or NF-kB signaling and production of inflammatory cytokines, as a potential mechanism contributing to inflammation in COVID-19 and PASC [15,18,19,31]. To test this hypothesis, human A549 Type 2 alveolar lung epithelial cells (stably overexpressing ACE2), as well as Caco-2 human intestinal epithelial cells (which express high levels of ACE2), were exposed to the S1 protein in cell culture and analyzed for cytokines in the supernatant as well as MAPK, ERK1/2, and NF-kB p-p65 activation by western blotting. Multiple organ systems are impacted by COVID-19 and PASC and well-documented effects are noted for the respiratory, cardiovascular, gastrointestinal, and central nervous systems [12,32]; the S1 spike protein might directly influence the effects in the lung, gastrointestinal tract, and brain [17,19,31].

## 2. Materials and Methods

### 2.1. Cell Culture

This study utilized two cell lines (A549 and Caco-2) that have both been used extensively in studies of infection with intact SARS and SARS-CoV-2 viruses [33,34,35].

A549 human lung Type 2 alveolar epithelial cancer cells from ATCC (#CCL-185, Bar Harbor, ME) were used in this study. This A549 human lung Type 2 epithelial cell line has relatively low expression of ACE2, and we opted to overexpress ACE2 to ensure a robust signal, an approach that has been used previously [34,35]. COVID-19 patients (both acute and Long COVID) exhibit respiratory symptoms and pathology, making evaluation of lung epithelial cells highly relevant [14,36].

Caco-2 human colon epithelial carcinoma cells from ATCC (#HTB-37, Bar Harbor, ME) were used in this study. Caco-2 cells express high levels of ACE2 protein and were used to isolate the first COVID-19 virus in Wuhan and to affirm ACE2 as the principal receptor for SARS-CoV-2 in early 2020 [34,37]. Intestinal epithelial cells are relevant because COVID-19 patients (both acute and PASC) exhibit gastrointestinal symptoms and pathology [14,36,38]. Caco-2 cells robustly express ACE2 and can be infected by SARS [39] and SARS-CoV-2 [34,35].

Cells were cultured in DMEM (Fisher Scientific, Waltham, MA, USA) supplemented with 10% FBS (Gibco, Waltham, MA, USA), 1 mM sodium pyruvate (Life Technologies, Waltham, MA, USA), 2 mM L-glutamine (Fisher Scientific), 10 mM HEPES (Fisher Scientific), 100 U/mL of penicillin, and 100 μg/mL of streptomycin (Fisher Scientific).

ACE2 stable overexpression in A549 cells (operationally designated as A549+ cells in this paper) was conducted, as previously described [40]. In brief, A549 cells were transfected with psPAX2, pCMV-VSVG, and pRRL.sin.cPPT.SFFV/Ace2.IRES-puro.WPRE (a gift from Caroline Goujon; Addgene plasmid #145839; RRID:Addgene145839, Addgene, Watertown, MA, USA). These A549 cells were subsequently transduced with the lentiviral particles and puromycin (InvivoGen, San Diego, CA, USA) that were selected as previously described [41]. ACE2 expression for A549, A549+ and Caco-2 cells was validated by western blotting (Figure 1).

### 2.2. Experimental Manipulation

A549+ cells and Caco-2 cells were grown to 80% confluence and cells were subsequently stimulated with either 10 ng/mL (0.13 nM), 25 ng/mL (0.325 nM), or 50 ng/mL (0.65 nM) of S1 protein per well in complete media for 6 h or 24 h. We chose to use the recombinant Raybiotech S1 protein in these experiments, which is the most widely used S1 spike [42,43,44], to allow clear comparison with other S1 studies. All experiments were run in triplicate or greater. For inhibition experiments, MEK1/2 inhibitor PD98059 (30 µM final) was used to pretreat the cells 1 h before the S1 protein was added (the inhibitor was not removed during S1 exposure). At the end of each experiment, media were collected for analysis of cytokines and cell lysates were collected for western blot analysis of signaling proteins.

SARS-CoV-2 Spike recombinant protein S1 subunit #230-01101-100, Wuhan WT S1 protein [45] was obtained from RayBiotech (Peachtree Corners, GA) (accession #QHD43416; mw 75 kD (Val16—Gln690): AA sequence: 16 vnltt rtqlppaytn sftrgvyypd kvfrssvlhs tqdlflpffs 61 nvtwfhaihv sgtngtkrfd npvlpfndgv yfasteksni irgwifgttl dsktqslliv 121 nnatnvvikv cefqfcndpf lgvyyhknnk swmesefrvy ssannctfey vsqpflmdle181 gkqgnfknlr efvfknidgy fkiyskhtpi nlvrdlpqgf saleplvdlp iginitrfqt 241 llalhrsylt pgdsssgwta gaaayyvgyl qprtfllkyn engtitdavd caldplsetk 301 ctlksftvek giyqtsnfrv qptesivrfp nitnlcpfge vfnatrfasv yawnrkrisn 361 cvadysvlyn sasfstfkcy gvsptklndl cftnvyadsf virgdevrqi apgqtgkiad 421 ynyklpddft gcviawnsnn ldskvggnyn ylyrlfrksn lkpferdist eiyqagstpc 481 ngvegfncyf plqsygfqpt ngvgyqpyrv vvlsfellha patvcgpkks tnlvknkcvn 541 fnfngltgtg vltesnkkfl pfqqfgrdia dttdavrdpq tleilditpc sfggvsvitp 601 gtntsnqvav lyqdvnctev pvaihadqlt ptwrvystgs nvfqtragcl igaehvnnsy 661 ecdipigagi casyqtqtns prrarsvasq 690). Chemical inhibitor for MEK1/2 PD98059 was obtained from Selleck Chemicals, LL (#S1177, Houston, TX, USA).

### 2.3. Western Blotting Analysis

Western blotting was performed, as previously described [40,46]. In brief, cells were lysed with Tris-triton buffer (Bioworld, Fisher, Pittsburg, PA, USA) with a phosphatase/protease inhibitor cocktail (Sigma, St. Louis, MO, USA). Total protein was determined (Bio-Rad, Hercules, CA, USA) and samples were prepared with Laemmli sample buffer with 2-ME (Bio-Rad), as previously described. Thirty micrograms of protein were loaded into each lane of a 4%/10% stacking acrylamide Tris gel and electrophoresed at 100 V for 2 h. Protein was transferred to a nitrocellulose membrane (GE Healthcare Limited, Buckinghamshire, UK) for 1.5 h at 130 V. Nonspecific binding was blocked by incubation of the membrane with 5% milk TBST for 1 h. Membranes were then incubated overnight at 4 °C with antibodies for p-ERK1/2, ERK1/2, NF-kB p-p65, NF-kB p65, hACE2, or h-actin in TBST and 5% nonfat dry milk. Membranes were washed with TBST for 1 h and subsequently incubated with the appropriate horseradish peroxidase (HRP)-conjugated anti-mouse/rabbit secondary antibody (below) for 1 h followed by washing with TBST for 1 h. Chemiluminescent substrate (ECL, GE Healthcare) was applied to the membrane for protein visualization using autoradiography film (HyBlot CL, Denville Scientific, Metuchen, NJ). Optical density was determined via densitometric analysis with ImageJ Software (NIH, Bethesda, MD, USA). After assessing p-ERK1/2 or NF-kB p-p65, blots were stripped and assessed for total ERK1/2 or NF-kB p65, respectively. Data were normalized to actin for each lane, for densitometry comparisons [46]. Antibodies were obtained from cell signaling technology (Danvers, MA, USA), including #7076S anti-mouse HRP; #7074 HRP-linked goat anti-rabbit IgG; #4370 phospho-p44/42 MAPK (ERK1/2) Thr202/Tyr204 rabbit mAb; #9102 total p44/42 MAPK (ERK1/2) rabbit poly from Millipore-Sigma (Burlington, MA); #A2066 anti-β-actin human rabbit poly from Novus Biologicals (Centennial CO); anti-ACE2 human/mouse, rabbit monoclonal: #NBP2-67692; from Santa Cruz Biotechnology (Dallas, TX, USA); anti-NFκB p65 antibody (F-6): #sc-8008, mouse monoclonal; and NF-kB p-p65 #sc-166748 mouse monoclonal.

### 2.4. Meso Scale Analysis of Cytokines

Analysis of cytokines in supernatants was carried out, as we have recently described, using the Meso Scale Diagnostics platform (Meso Scale Diagnostics LLC, Rockville, MD, USA) [47]. This analysis utilized the Meso Scale V-Plex Proinflammatory Panel 1 Human Kit (# K15049D), which evaluates 10 cytokines: IFN-γ, IL-1β, IL-2, IL-4, IL-6, IL-8, IL-10, IL-12p70, IL-13, and TNF-α. (IFN-γ range: 0.37–938 pg/mL; IL-1β range: 0.05–375 pg/mL; IL-2 range: 0.09–938 pg/mL; IL-4 range: 0.02–158 pg/mL; IL-6 range: 0.06–488 pg/mL; IL-8 range: 0.07–375 pg/mL; IL-10 range: 0.04–233 pg/mL; IL-12p70 range: 0.11–315 pg/mL; IL-13 range: 0.24–353 pg/mL; and TNF-α range: 0.04–248 pg/mL). Supernatant (25 µL) from each well was combined with 25 µL Meso Scale Diagnostics test solution, processed, and evaluated in the Meso Scale Diagnostic instrument: MESO QuickPlex SQ 120 MM according to manufacturer’s instructions. Media alone were used as a control group, an approach used in other published S1 spike studies. [43,48,49].

Sample values below the level of detection (LOD) of the Meso Scale assay were set to the lowest detectable level of the assay for that cytokine when used to calculate means. If all values measured for a cytokine were below the LOD (IFN-γ, IL-2, IL-4, IL-10, or IL-12p70), these cytokines’ data were not analyzed, nor are the data shown. Specifically, IL-1β, Il-13, Il-6, IL-8, and TNF-α were detectable in a majority of A549+ cell supernatant samples after S1 stimulation and, therefore, are included in Figure 2. In Figure 4, only IL-1β, IL-6, and IL-8 were detectable in most supernatants from Caco-2 cells after S1 stimulation and, therefore, they are included for analysis. No data showed a detectable level for A549+ or Caco-2 cells supernatants for IFN-γ, IL-2, IL-4, IL-10, or IL-12p70, the data for which were therefore omitted.

### 2.5. Statistical Analysis

Based on the Shapiro–Wilks normality test, statistical analyses were carried out using either the Student’s *t*-test, the Mann–Whitney U (MWU) test, or one-way analysis of variance (ANOVA) with a priori between-group testing using the Tukey’s multiple comparisons test. All analyses were conducting using GraphPad Prism (v9.3.1) (GraphPad Software LLC, San Diego, CA, USA) with significance set to (*p* < 0.05). Densitometry and scanning of western blotting data was carried out with ImageJ (National Institutes of Health software).

## 3. Results

### 3.1. Expression of SARS-CoV-2 Receptor ACE2 in A549, A549+, and Caco-2 Cell Lines

ACE2 is the principal receptor for SARS-CoV-2 binding to human cells [34,50]. The native ATCC cell line human lung Type 2 alveolar A549 cells express low levels of ACE2 protein (Figure 1a, lanes 1–2). This finding mirrors native A549 cells and primary Type 2 alveolar cells that express ACE2, but not at high levels, although they robustly support SARS and SARS-CoV-2 infection [34,35]. However, to optimally assess the study hypothesis, ACE2 was overexpressed in A549 cells using a protocol that our group previously used to stably overexpress ACE2 in HeLa cells [40]. This is a strategy used by others in COVID-19 studies [34,35]. These cells were operationally defined as A549+ and robustly express ACE2 receptor protein (Figure 1a, lanes 3, 4, and 8). They were used for all subsequent experiments. The Caco-2 cells used in this study robustly express the ACE2 protein (Figure 1a, lanes 5–7). Multiple lanes are shown to demonstrate reproducibility. These western blot densitometry data are summarized in Figure 1b, one-way ANOVA: F _(2,14)_ = 410.6, *p* ˂ 0.001), and serve as the rationale for the use of the A549+ and Caco-2 cells in these subsequent experiments. Thus, the cells used to assess the study’s hypothesis in our system modeling the cytokine and signaling effects of S1 protein binding to human cells expressing ACE2 each express robust levels of ACE2, as described by others [35].

### 3.2. The SARS-CoV-2 S1 Spike Protein Stimulates Production of the Key COVID-19 Inflammatory Cytokine IL-1β in Human A549+ Lung Cells That Is Blocked with the MEK1/2 MAPK ERK1/2 Inhibitor

Studies in human lung vascular cells show that S1 spike treatment activates MAPK ERK1/2, which can be blocked with an MEK1/2 inhibitor [43]. Another recent study with live SARS-CoV-2 virus and A549 cells showed that live SARS-CoV-2 virus stimulates inflammatory cytokines that could be blocked with an MEK1/2 (MAPK ERK1/2) inhibitor [35]. Therefore, we sought to determine whether the S1 spike protein alone could stimulate cytokine production in A549+ cells and if that effect could be blocked with the MEK1/2 inhibitor PD98059 (see Section 2. A549+ cells were treated with either media control or 50 ng/mL S1 protein alone or S1 protein with MEK1/2 inhibitor (30 µM final) for 24 h. Inhibitor was introduced 1 h before S1 stimulation and remained. Cytokines in culture supernatants were quantified using the Meso Scale (see Section 2).

Supernatants were subsequently analyzed for key inflammatory cytokines associated with the cytokine storm in COVID-19 (IFN-γ, IL-1β, IL-2, IL-4, IL-6, IL-8, IL-10, IL-12p70, IL-13, and TNF-α) [9,51,52,53], using the Meso Scale platform. Only a subset of five cytokines tested resulted in measurable values; these are shown in Figure 2, including IL-1β, IL-13, IL-6, IL-8, and TNF-α. Values for IFN-γ, IL-2, IL-4, IL-10, and IL-12p70 were below the level of detection (LOD) of the assay and were not analyzed. Post hoc analysis indicated that S1 spike treatment (50 ng/mL for 24 h) significantly increased supernatant IL-1β (Tukey’s: *p* ˂ 0.05), an effect that was significantly inhibited by the MEK1/2 ERK1/2 inhibitor (Tukey’s: *p* ˂ 0.01) (Figure 2a). No other cytokines were increased by S1 treatment; however, post hoc analyses for cytokine production was significantly inhibited by the MEK1/2 inhibitor for IL-13 (Tukey’s: *p* ˂ 0.0001), IL-8 (Tukey’s: *p* ˂ 0.0001), and TNF-α (Tukey’s: *p* ˂ 0.05) (Figure 2b–d). MEK/MAPK signaling often drives baseline or the early expression of many cytokines and MEK/MAPK inhibitors are widely used as anti-inflammatory cytokine inhibitors, including usage in recent COVID-19 studies [35,51]. No differences were noted in IL-6 production under any condition (Figure 2e). These data support that conclusion the S1 spike protein is sufficient to induce the production of the key COVID-19 pro-inflammatory cytokine IL-1β via a MEK1/2 dependent MAPK-ERK 1/2 pathway in A549+ human lung epithelial cells, a major COVID-19 cell-type target [38].

### 3.3. The SARS-CoV-2 S1 Spike Protein Activates the MAPK-ERK1/2 and the NF-kB p65 Signaling Pathways in Human A549+ Lung Cells

SARS-CoV-2 spike S1 protein and live SARS-CoV-2 virus have each been shown to activate MEK/MAPK signaling in lung vascular cells [35,43]. SARS-CoV-2 infection of A549 cells with live SARS-CoV-2 virus in a recent study resulted in activation of MAPK ERK1/2 signaling [35]. Those findings are expanded in our data in Figure 2a, with the novel finding that a MEK1/2 MAPK ERK1/2 inhibitor blocks the S1 spike-stimulated production of IL-1β in A549+ lung epithelial cells [43]. In addition, the same study showed that SARS-CoV-2 virus infection of A549 cells stimulates NF-kB p65 activation [35]. Other recent studies have shown that S1 spike can activate NF-kB p65 in A549 cells [49]. Therefore, we next sought to determine whether the S1 spike protein alone was sufficient to stimulate MAPK ERK1/2 (p-ERK1/2) or NF-kB (p-p65) activation in our A549+ lung cells, using western blotting of A549+ cell lysates to measure phosphor-ERK1/2 MAPK (activated) as well as NF-kB phosphor-p65 (activated). We compared the treatment of A549+ cells with the S1 protein 25 ng/mL or 50 ng/mL vs. media control for 24 h and measured the MAPK pathway activation (i.e., p-ERK1/2) by western blotting. Post hoc analysis found that a significant p-ERK1/2 increase was noted with 50 ng/mL S1 spike protein, but not with 25 ng/mL S1 (Tukey’s: *p* ˂ 0.0181) (Figure 3a). This effect on p-ERK1/2 activation with 50 ng/mL S1 was blocked by pre-treatment of A549+ cells with the MEK1/2 MAPK ERK1/2 inhibitor (PD98059 30µM) (Tukey’s: *p* ˂ 0.0072) (Figure 3b). Our post hoc data also showed that treatment of A549+ cells with S1 spike resulted in significant activation of NF-kB p-p65 with 50 ng/mL S1 protein (Tukey’s: [control vs. S1-50 ng] *p* ˂ 0.0336; [S1-25 ng vs. S1-50 ng] *p* ˂ 0.0247), but not with 25 ng/mL S1 protein (Figure 3c). However, in a second round of experiments stimulating with spike S1 50 ng/mL with/without the MEK1/2 inhibitor, we found no effect of the MEK1/2 inhibitor on S1 spike protein mediated NF-kB p-p65 activation, because the S1 spike protein alone did not stimulate NF-kB p-p65 activation in A549+ cells (Figure 3d). It is not clear why Figure 3c–d data for S1 stimulation of NF-kB p-p65 differed for 50 ng/mL S1, but our data for IL-1β activation in Figure 2a support NLRP3 inflammasome/NF-kB (to activate IL-1β) activation by 50 ng/mL S1 spike [54,55]. In addition, other recent published S1 spike data support our data in Figure 3b with demonstration that the S1 protein stimulates NF-kB in A549 human lung cells [49,56], supporting S1 activation of the NF-kB pathway in A549+ cells. Our rationale for testing the MAPK inhibitor effects on NF-kB activation by S1 spike was that one recent report supported the view that MEK1/2 MAPK inhibition could block the SARS-CoV-2 virus activation of NF-kB in lung epithelial cells [35].

### 3.4. The SARS-CoV-2 S1 Spike Protein Stimulates Inflammatory Cytokines IL-6 and IL-8 Production in Caco-2 Human Intestinal Epithelial Cells

The S1 spike protein had distinct effects on A549+ cells, and next we evaluated the effects of the S1 spike protein stimulation on cytokine production by Caco-2 human intestinal epithelial cells. As a preliminary experiment, a dose-response test was conducted in Caco-2 cells, in which cells were grown in cell culture and stimulated for 24 h with S1 spike concentrations of 10 ng/mL, 25 ng/mL, or 50 ng/mL S1 spike protein (N = 3 each), and cytokine production in supernatants were assessed. Supernatants were tested with the Meso Scale panel for 10 cytokines as for A549+ cells (see Section 2). Only data for stimulation with 50 ng/mL are shown in Figure 4 (10 ng/mL and 25 ng/mL data were not significant, and are not shown). While the S1 spike protein did not significantly increase IL-1β production by Caco-2 cells (Figure 4a, MWU: *p* = 0.4000), contrary to what was observed in A549+ cells in Figure 2a, S1 did significantly increase IL-6 (Figure 4b, *t*-test: *p* < 0.0048) and IL-8 (Figure 4c, *t*-test: *p* < 0.0204) production at 24 h by Caco-2 cells. This finding is highly relevant, as both IL-6 and IL-8 are core members of the inflammatory cytokine storm in COVID-19 [9,11]. No effects of S1 spike stimulation of Caco-2 cells were noted for IFN-γ, IL-13, IL-2, IL-10, IL-12p70, IL-13, IL-4, and TNF-α, which were below the level of detection of the assay. 

### 3.5. The SARS-CoV-2 S1 Spike Protein Does Not Stimulate MAPK p-ERK1/2 or NF-kB p-p65 Signaling Pathways in Human Caco-2 Intestinal Epithelial Cells

In our final set of experiments, we evaluated the effects of S1 spike stimulation on Caco-2 intestinal cell ERK1/2 MAPK and NF-kB p65 signaling at 24 h, using western blotting as shown in Figure 5 (see Section 2). In Figure 5a, Caco-2 cells were stimulated for 24 h with either 25 ng/mL or 50 ng/mL S1 spike protein or media control; p-ERK1/2 activation (phosphorylation) was measured by western blotting. No significant (ns; *p* > 0.05) effect of S1 spike treatment on p-ERK1/2 activation was observed in Caco-2 cells (Figure 5a), which was replicated in six independent experiments and the MEK1/2 inhibitor had no effect. Caco-2 cells were next stimulated for 24 h with media control or 50 ng/mL S1 spike alone, or 50 ng/mL S1 spike with MEK1/2 inhibitor added 1 h before treatment and remaining. Again, no effect of S1 treatment on ERK1/2 activation or by the MEK1/2 inhibitor was observed (Figure 5b, ns; *p* > 0.05). Next, we assessed NF-kB p-p65 phosphorylation (activation) by S1 spike treatment in Caco-2 intestinal cells. As shown in Figure 5c, Caco-2 cells were stimulated with media control or 25 ng/mL S1 or 50 ng/mL S1 spike. Data showed that no significant effect of either S1 spike concentration was observed on NF-kB p-p65 phosphorylation at 24 h. Finally, Figure 5d data summarize the stimulation of Caco-2 cells with media control or 50 ng/mL S1 spike protein alone or 50 ng/mL S1 spike plus the MEK1/2 inhibitor. These data also supported the conclusion that there was no effect of S1 spike protein treatment on NF-kB p-p65 activation and no effect of the MEK1/2 inhibitor on the NF-kB p-p65 activation. In summary, we found no statistically significant effects of S1 spike protein stimulation on Caco-2 intestinal epithelial cell p-ERK1/2 MAPK activation or NF-kB p-p65 activation signaling in Caco-2 intestinal epithelial cells, which suggests that the observed increase in IL-6 and IL-8 (Figure 4b–c) probably occurs through signaling pathways independent of these pathways, which is also a novel finding.

## 4. Discussion

In summary, the data from these experiments support the following conclusions. The stimulation of A549+ human lung Type 2 alveolar cells with SARS-CoV-2 S1 spike protein (50 ng/mL, 24 h) results in significantly increased supernatant IL-1β, which could be blocked with a MEK1/2 inhibitor for ERK1/2 MAPK. In addition, while the S1 spike protein did not augment the production of IL-13, IL-8, and TNF-α, the MEK1/2 ERK1/2 inhibitor reduced the baseline expression of cytokines IL-13, IL-8, and TNF-α in A549+ cells. However, no significant effect of S1 spike stimulation on these three cytokines or IL-6 increase was observed in A549+ cells (Figure 2). Western blotting analysis of A549+ cells for p-ERK1/2 MAPK activation and NF-kB p-p65 activation revealed that 50 ng/mL S1 spike protein stimulated both ERK1/2 activation and NF-kB p-65 activation at 24 h. The MEK1/2 ERK1/2 MAPK inhibitor blocked the effect of S1 spike protein on the activation of p-ERK1/2 MAPK. However, the effects of the MEK1/2 inhibitor on NF-kB p-p-65 activation were not significant. (However, additional studies are necessary to confirm this finding, due to between experiment variability (Figure 3)). A different profile was observed following stimulation of Caco-2 human intestinal epithelial cells with S1 spike protein. The S1 spike protein (50 ng/mL for 24 h) did not significantly increase IL-1β release as in A549+ cells (Figure 2a), but did significantly increase the supernatant production of the key inflammatory cytokines IL-6 and IL-8 in human Caco-2 cells (Figure 4). Using western blotting, we found the increase in IL-6 and IL-8 occurred independent of the stimulation of Caco-2 intestinal cells p-ERK1/2 activation or NF-kB p-p65 activation, which were unaffected by the S1 spike protein vs. media controls. Caco-2 cells stimulation with 25 ng/mL or 50 ng/mL S1 spike protein for 24 h did not result in any significant p-ERK1/2 or p-p65 NF-kB activation and the MEK1/2 inhibitor had no measurable effect on measures of p-ERK1/2 MAPK or NF-kB p-p65 activation after treatment with 50 ng/mL S1 spike for 24 in Caco-2 human intestinal cells (Figure 5).

Taken together, the results from these experiments demonstrate cell-type-specific changes in the response of human lung epithelial A549+ cells and Caco-2 human intestinal epithelial cells to stimulation with the SARS-CoV-2 S1 spike protein. One key finding is the S1 spike protein-induced production of IL-1β production that was blocked with the MEK1/2 MAPK inhibitor for ERK1/2 in A549+ cells (Figure 2a). This has important implications for the hypothesis that the S1 spike protein may contribute to pathogenicity in both COVID-19 and PASC because IL-1β is a primary cytokine candidate to drive the COVID-19 cytokine storm and downstream cytokine production [11,57]. Treatments that target IL-1β production early in COVID-19 infection block the cytokine storm [58,59]. No other cytokines in our ten-cytokine panel were stimulated by the S1 spike in A549+ lung cells. In addition, the MEK1/2 MAPK ERK1/2 inhibitor significantly reduced the baseline expression of the three cytokines IL-13, IL-8, and TNF-α by the A549+ cells. However, no significant effect of S1 spike stimulation of these three cytokines or IL-6 was observed. In contrast, the stimulation of Caco-2 human intestinal epithelial cells with the same dose of S1 spike (50 ng/mL, 24 h) did not significantly increase IL-1β as in A549+ cells (Figure 2a), but did significantly increase the production of the inflammatory cytokines IL-6 and IL-8, which are also members of the cytokine storm (Figure 4) [9]. We also examined the activation of the ERK1/2 MAPK and NF-kB p65 signaling pathways using western blotting and the effects of an MEK1/2 inhibitor on S1 stimulation of these pathways in A549+ as well as Caco-2 cells. The rationale for investigating the MEK/12 MAPK inhibitor on S1 signaling was that several studies have shown that SARS-CoV-2 infection, as well as S1 spike, activates MAPK ERK1/2 and that MEK1/2 inhibitors or siRNA knockdown of ERK1/2 can block early SARS-CoV-2 virus or S1 spike signaling in different cell types [18,35,43]. In this study, S1 spike treatment was sufficient to activate the MAPK signaling pathway (ERK1/2) in A549+ cells, whereas these pathways were unaffected in Caco-2 cells [18,19,43]. Although potentially from different organs, each organ (i.e., lung and intestine) could contribute to the systemic measurement of these cytokines during COVID-19 and *PACS*. Perhaps most importantly, this study demonstrated the inflammatory cytokine production of IL-6 and IL-8 by Caco-2 cells in response to the S1 spike protein, as well as a potentially different pathway than ERK1/2 or NF-kB p65, as our data showed no activation by S1 spike of these pathways. This is potentially significant because IL-6, especially, is a major target for therapy in COVID-19 [60].

A growing number of studies (in multiple cell types) demonstrated that treatment of cells with the S1 spike protein changes cell signaling, especially the activation of MAPK ERK1/2, and promotes pro-inflammatory cytokine production [18,19,43,61]. These largely lung cell types also include direct activation of monocytes and microglia by S1 spike [62,63]. In addition, tracheal gavage of the S1 spike protein in mice promotes lung inflammation [44]. Study-specific differences were noted in signaling and cytokine production in these studies, but they shared the theme of S1 spike protein-induced inflammation, thereby supporting the data in this report. The data in this study showed significant S1 spike stimulation of production of IL-1β at 24 h in A549+ cells, while Caco-2 cells showed significant production of IL-6 and IL-8 at 24 h. These data were remarkable for several reasons. First, the collective profile of cytokines produced matches three core cytokines of the COVID-19 cytokine storm, IL-1β, IL-6, and IL-8, with no anti-inflammatory cytokines produced [9,11]. Perhaps most importantly, this study demonstrated inflammatory cytokine production by Caco-2 cells in response to the S1 spike protein. To date, only a single study evaluated the impact of the spike protein on Caco-2 cells; it found that the entire spike protein (i.e., the S1-S2 complex at 24 h) promotes the production of inflammatory cytokines IL-β, TNF-α, IL-6, and IL-18 via a PPARγ mechanism, not MAPK, which is consistent with our results [64]. Two recent studies noted that while Caco-2 cells permit robust infection by the COVID-19 virus, the cells exhibit no cell death or other pathology [65,66]. Thus, the Caco-2 intestinal cells’ response to SARS-CoV-2 virus appears to be quite different from that of lung epithelial A549+ cells [65]. Our study observed a significant stimulation of MAPK ERK1/2 by the S1 spike protein in A549+ lung cells. This effect of the S1 spike protein was also noted to activate these pathways in other cell types, including platelets, lung vascular cells, and A549 cells [18,26,43,56,67]. The MEK1/2 inhibitor blocked the S1-simulated increase in ERK1/2 MAPK phosphorylation (activation) (Figure 3b), which replicates other recent reports [18,43]. No effects of the S1 spike protein were noted on the MAPK pathway in Caco-2 cells. According to the relevant literature, the S1 spike protein stimulates NF-kB p-p65 in A549 cells and, in other studies, blocking NF-kB prevents the observed increase in cytokines [49,68]. In addition, the entire (i.e., S1-S2 complex) spike protein activates MAPK and NF-kB pathways as well as cytokine production in A549 cells, an effect that is blocked with MAPK and NF-kB inhibitors [69]. A recent study also demonstrated that a MAPK inhibitor blocks the SARS-CoV-2-induced production of inflammatory cytokines in a humanized COVID-19 mouse model, wherein the SARS-CoV-2 virus transiently activated MAPK signaling and knockdown of this pathway (i.e., ERK1/2) and prevented inflammatory cytokine production (i.e., IL-6, TNF-α) in these mice [35]. However, in contrast, the S1 spike protein did not promote an increase in MAPK or NF-KB signaling in these pathways in Caco-2 cells. This suggests that the S1-mediated production of cytokines in Caco-2 cells may occur independently of the MEK/MAPK pathway. The data in this report support the conclusion that the S1 spike protein may promote the production of cytokines via the MAPK pathway, at least in A529+ cells. In light of these recent data for MAPK inhibitors in S1 and COVID-19 models, as noted above, MAPK inhibition could be a possible treatment target for COVID-19 and *PACS*, as has been suggested by others [35].

Taken together, these data are important in supporting the growing body of studies that show that the S1 spike protein has proinflammatory biological activity and may contribute to COVID-19 and PASC [18,19,31]. Several variants of concern (VOC) have emerged since the original Wuhan SARS-CoV-2 strain, and these are largely distinguished by mutations in the spike protein, especially in the S1 protein [70,71]. There is evidence to suggest that the S1 spike protein in VOC may be associated with an altered inflammatory profile [71]. A mutation in the Delta S1 spike protein that promotes increased pathogenicity may also drive enhanced production of TNF-α, IL-6, and IL-1β [72,73]. Mutations in the spike protein (37 in spike total, 26 in S1) have also been noted in the Omicron variant [74,75,76], which appears to be associated with less NF-kB activation compared with other variants but with a 2.5-fold increase in ACE2 affinity [71,77]. Studies have also shown that recombinant Omicron S1 protein induces only weak humoral and cellular immunity in mice that may be associated with less NF-kB activation [78]. Clearly, this is an evolving area and each mutated spike protein in VOC have distinct effects in terms of pathway activation and cytokines produced. The differences may relate to affinity of the spike protein for ACE2 or other nuances that have yet to be defined.

There is growing evidence supporting a role for the S1 spike protein in PASC [18,19,31]. Earlier studies supported a sequence motif in the SARS-CoV-2 S1 protein, with high similarity to bacterial super antigens that stimulate persistent cytokine production, similar to that observed in COVID-19 [79]. Recent studies have supported the model that persistent systemic circulating levels of the SARS-CoV-2 S1 spike protein are associated with PASC [80,81]. S1 protein injection is sufficient to induce neuroinflammation and production of cytokines in rats [82]. One recent study demonstrated that S1 spike protein is present in monocytes of PASC patients as long as 15 months after initially testing SARS-CoV-2 positive [80]. A recent preprint demonstrated that spike protein is found in the blood of a majority of PASC patients up to 12 months post-diagnosis, which was not observed in early recovered COVID-19 patients [83]. These studies support the presence of a persistent spike protein reservoir in the blood of the majority of PASC patients but not in early COVID-19 patients. Finally, a recent study of PASC patients supported an immunity model wherein there was a prolonged immune response profile associated with the spike protein in *PASC* patients [81]. More studies are needed to better understand the role of the S1 spike protein in COVID-19 and PASC [19,83].

There are limitations to this study that should be noted. Some studies with the Raybiotech S1 spike used in our study found only transient activation of MEK (which activates ERK1/2) in human lung vascular endothelial and smooth muscle cells. However, these are different cell types. In our discussion, we also point out that there is a great deal of variability in the published data regarding S1 activation of signaling pathways and cytokine production, which may be related to concentrations of the S1 protein and the time course. Future studies are needed and must include additional cell types, a larger dose–response range, an extended time–course evaluation, and an evaluation of the effects of different spike proteins from Omicron and other SARS-CoV-2 VOCs. It is noteworthy that this current study used a lower concentration of the S1 spike protein than the concentrations reported in other studies [44,84]. However, despite these limitations, this study supports the concept that the SARS-CoV-2 S1 spike protein can alter MAPK signaling pathways and promote COVID-19-characteristic cytokine production in relevant human lung epithelial and intestinal epithelial cells that may contribute to COVID-19 cytokine storm pathology, as well as possibly contributing to *PACS* pathology.

## 5. Conclusions

In conclusion, data in this report demonstrated that the SARS-CoV-2 S1 spike protein activates cytokine production, ERK1/2 MAPK activation, and NF-KB p65 activation in a cell-type specific manner. The S1 spike protein stimulated the production of cytokines: in human lung A549+ cells, we found that S1 spike stimulated the production of the key COVID-19 cytokine IL-1β, and in human intestinal epithelial Caco-2 cells, S1 stimulated the production of IL-6 and IL-8 cytokines. Importantly, we also found that IL-1β in A549+ human lung cell supernatants was blocked with an MEK1/2 ERK1/2 MAPK inhibitor. We also demonstrated, for the first time, that the SARS-CoV-2 S1 spike stimulates the Caco-2 human intestinal epithelial cell production of IL-6 and IL-8 in cell supernatants. These cytokines are all core features of the cytokine storm observed in COVID-19 infection [9]. Taken together, these data support a possible role for the S1 spike protein in COVID-19 and *PASC* inflammation pathogenesis and, potentially, MAPK inhibitors as therapies for COVID-19 or *PASC*, as has been suggested by others [19,31,35]. In addition, as others have pointed out, the biological effects of the S1 protein could have implications for side effects that were observed following the administration of the COVID-19 mRNA and DNA vaccines that promote the body to synthesize the S1-S2 spike protein [18]. 

## Figures and Tables

**Figure 1 microorganisms-10-01996-f001:**
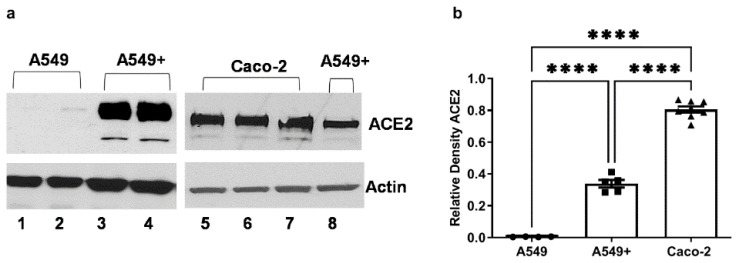
ACE2 receptor protein expression in A549, A549+, and Caco-2 cells in this study. The three cell lines utilized in this study were evaluated for ACE2 receptor protein expression using western blotting. A549 and A549+ human Type 2 lung alveolar cells and Caco-2 human intestinal epithelial cells were grown to 80% confluence in 12-well plates, and each lane (30 µg/lane) represented a separate well. Cells were lysed and analyzed with antibody to human ACE2 and densitometry analysis was conducted using ImageJ software (NIH). (**a**) Lanes 1–2 are native ATCC A549 cells; lanes 3–4 are A549+ ACE2 stable overexpressing cells; lanes 5–7 are lysates from human Caco-2 intestinal epithelial cells; lane 8 shows data from a single well of A549+ cells, with Caco-2 cell lysates for comparison on the same blot. (**b**) Relative ACE2 western blot densitometry data comparisons (each data point is mean ACE2 from a well) for the three cell types. A549, N = 4; A549+, N = 5; Caco-2, N = 8. The actual ImageJ mean density values for relative ACE2 protein (~120 kD) expression vs. actin (~45 kD) are for A549 mean 0.004207; A549+ mean 0.3384, and Caco-2 cells mean 0.8054. One-way ANOVA (F _(2,14)_ = 410.6, *p* ˂ 0.001) was followed by Tukey’s multiple comparison test, as shown on the graph: **** *p* < 0.0001.

**Figure 2 microorganisms-10-01996-f002:**
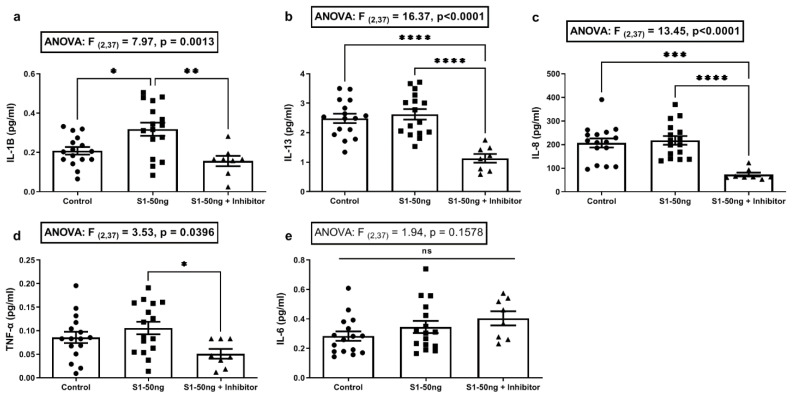
Supernatant cytokine levels following stimulation of A549+ cells with 50 ng/mL S1 spike protein for 24 h. A549+ cells in 12-well plates at 80% confluence in 1 mL media were stimulated for 24 h with media alone (“Control”, left column), or 50 ng/mL (0.65 nM) of the SARS-CoV-2 spike S1 protein (“S1-50 ng”, middle column), or 50 ng/mL S1 plus 30µM PD98059 MEK1/2 MAPK ERK1/2 inhibitor for 24 h (“S1-50 ng + Inhibitor”, right column). Inhibitor was added 1 h before S1 stimulation and was not removed. Supernatant cytokines were measured using the Meso Scale platform (**a**–**e**). Control: N = 16/group, S1-50 ng: N = 16/group, S1-50 ng + Inhibitor: N = 8/group). One-way ANOVA results (indicated in box) along with Tukey’s multiple comparison tests are shown on the graph: * *p* < 0.05, ** *p* < 0.01, *** *p* < 0.001, **** *p* < 0.0001, ns = not significant (*p* > 0.05).

**Figure 3 microorganisms-10-01996-f003:**
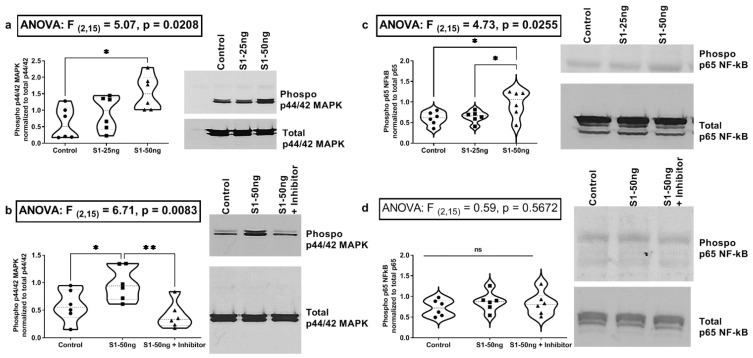
Western blotting data for A549+ human lung cells p-ERK1/2 MAPK and NF-kB p-p65 activation after 24 h stimulation with S1 spike +/− MEK1/2 (ERK1/2 MAPK) inhibitor. (**a**) A549+ cells were stimulated for 24 h with either media control (control) or S1 spike protein 25 ng/mL (S1-25 ng) or 50 ng/mL (S1-50 ng). (**b**) S1 spike protein (S1-50 ng/mL) or S1 spike protein plus MEK inhibitor (“S1-50 ng + Inhibitor”, 30 µM PD98059) for 24 h. (**c**) Cells were stimulated as in (**a**). and analyzed for (**c**,**d**) NF-kB p-p65 (activated) vs. total NF-kB (~65 kD). (**d**) Cells were treated as in (**b**) and analyzed for NF-kB p-p65 vs. NF-kB p65 (N = 6, group). For (**a**–**d**), western blot analysis mean densitometry is graphed, with representative blots at right. One-way ANOVA results (indicated in box) along with Tukey’s multiple comparison tests are shown on the graph: * *p* < 0.05, ** *p* < 0.01, ns = not significant (*p* > 0.05). (N = 6, each condition). (See Section 2). Cell lysates were analyzed by densitometry (ImageJ, NIH).

**Figure 4 microorganisms-10-01996-f004:**
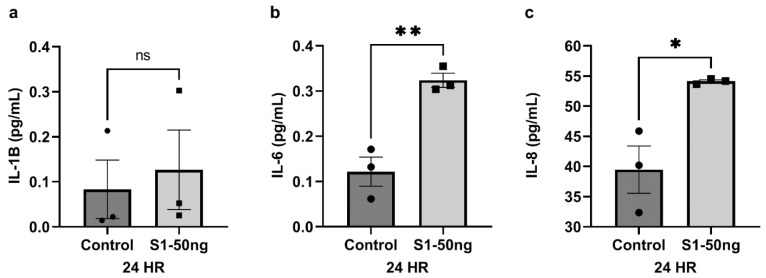
Supernatant cytokine levels following stimulation of Caco-2 human intestinal epithelial cells with S1 spike protein for 24 h. Caco-2 human intestinal epithelial cells grown in triplicate 12-well plates at 80% confluence in 1 mL media were stimulated with media alone (“Control”, left column) or 50 ng/mL SARS-CoV-2 S1 spike protein (“S1-50 ng”, right column) for 24 h. Cytokine levels in the supernatant were measured using the Meso Scale platform (**a**–**c**). N = 3/group. Mann–Whitney U test: ns = not significant (*p* > 0.05). Student’s *t*-test: * *p* < 0.05, ** *p* < 0.01.

**Figure 5 microorganisms-10-01996-f005:**
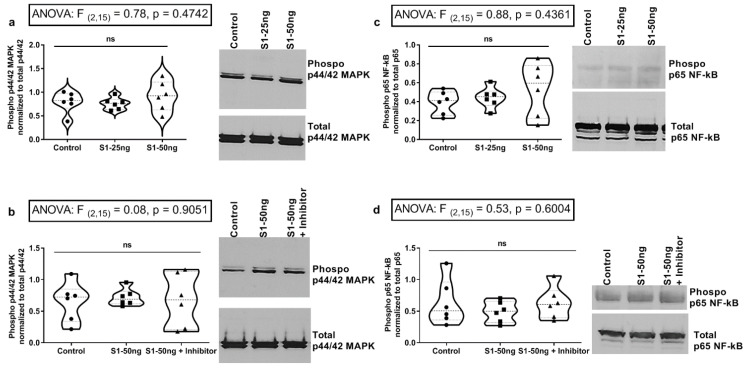
Western blotting analysis of Caco-2 human intestinal cells shows S1 spike protein does not activate p-ERK1/2 or NF-kB p-p65 signaling in stimulation with 24 h S1 Spike and +/− MEK1/2 inhibitor treatments. (N = 6 each condition, representative blots at right). (**a**) Caco-2 cells were stimulated for 24 h with either media control (Control), or 25 ng/mL S1 spike (S1-25 ng) or 50 ng/mL S1 spike (S1-50 ng) and cell lysate western blots were analyzed by densitometry and graphed. MAPK p-ERK1/2 (activated) vs. total ERK1/2 ratios were as described in Figure 3. (**b**) Caco-2 cells were treated with only media control or 50 ng/mL S1 spike or 50 ng/mL S1 spike plus 30 µM MAPK inhibitor PD98059 for 1 h before S1 stimulation and inhibitor was not removed for the 24 h. Cell lysates were analyzed as in (**a**) for p-ERK1/2 vs. total ERK1/2 (42/44 kD). (**c**) Caco-2 cells were stimulated for 24 h with either media control (left) or 25 ng/mL S1 spike or 50 ng/mL S1 spike (as in 5a/5b). Cell lysates were analyzed for NF-kB p-p65 (activated) and total NF-kB ratios by densitometry as in 3c. (**d**) Cells were treated and analyzed as in 3d above. NF-kB p-p65 (activated) vs. total NF-kB p65 (~65 kD) ratios were assessed. Densitometry ImageJ (NIH); one-way ANOVA results (indicated in box) along with Tukey’s multiple comparison tests are shown on the graph: ns = not significant (*p* > 0.05).

## Data Availability

No public archived databases were utilized in this study. Any requests for specific data can be directed to christopher_forsyth@rush.edu.

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
