# Peer review of "The SARS-CoV-2 S1 Spike Protein Promotes MAPK and NF-kB Activation in Human Lung Cells and Inflammatory Cytokine Production in Human Lung and Intestinal Epithelial Cells"

_microorganisms, 2022, doi:10.3390/microorganisms10101996_

Round 1

Reviewer 1 Report

SARS-CoV-2 is the biggest viral threat of our times. As such, it is admirable that scientists are exploring every aspect of the virus. For this, the authors must be congratulated. The manuscript describes the response to two cell lines to the addition S1 protein. This is intriguing given that S1 is cleaved upon viral entry into the cell and could thus act as a stimulant to other cells.

Whilst it is largely well written, my main criticism of this manuscript is that it over-extends when it comes to conclusions. Notably, whilst between them they might, neither of the cell lines really show a strong, robust, statistically significant upregulation of all of the cytokines the authors refer to. Arguably this even extends to the title; Fig 6 would suggest that there is no promotion of MAPK and NF-kB in Caco-2 cells by S1, yet the title implies that it does. In reality I think the results are more subtle, with the different cell lines showing different cytokine profiles.

Below is a list of points which I feel the authors should address prior to acceptance of the manuscript.

-From an international perspective I don’t think it’s accurate to state that there are only 3 main vaccines; it would be broader with respect to this statement.

-What about other S1 proteins? Having stated that S1 is frequently mutated in VOCs, would the authors anticipate that the S1 of different variants would behave similarly? Have they tried any?

-Related to the above point, the authors make a point of being very clear that the S1 protein is from RayBiotech. Why is this? Have other S1 proteins been tried without observing a phenotype?

-The sentence beginning on line 78 makes little sentence given the preceding sentence. I think it needs to be rephrased to make it clearer that the presence (or absence) of the cleavage site relates to pathogenesis.

-Convention would dictate that the acronym follows the complete phrase; as such the description of PAMP on line 108 should be reversed, i.e. to read ‘pathogen-associated molecular pattern (PAMP)‘.

-The sentence beginning “Summarised” on line 211 should be deleted. These details are provided in the legend.

-Why are there several lanes of seemingly the same sample in Fig 1a and b? 4/8 lanes of the same sample simply adds confusion. In reality this figure could simply comprise 3 lanes.

-Panel Fig 1b should be cropped on the right hand side; currently it appears that there should be a sample here.

-The legend of Fig 1a reads ‘lanes 5-7’; should this be 5-8?

-As a general point, I am a little unconvinced by the removal of results simply because they are zero. This does not seem robust to me. I suggest that a statistician verify that this is a valid approach. This is particularly evident in Figure 4, where, presumably as a result of removing lots of zero values, many of the statistical tests are not significant, e.g. only 4 points rather than 12 for panel Fig 4a.

-It would strengthen the case if a 'control + inhibitor' control was used in Fig 4. The decrease in Fig 4b might be statistically significant, but would the pattern change if the values for the control cells also dropped?

-I think the data would be strengthened by the addition of positive controls. I have not come across the methodology using Meso diagnostics to measure multiple cytokines, although this in itself is not a reason to be sceptical. However, without some form of positive control it is difficult to get an idea of what a positive reaction ‘looks like’. Some of the values seem rather small. This might be the case even with a positive stimulation, but without the control it’s impossible to say.

-The justification for only proceeding with A549 cells (for the data in Fig 4, lines 271-273) seems a bit obscure to me, particularly as Caco-2 cells re-enter the story later on. Could the authors please clarify this a bit explicitly.

-Have the authors included the sentence beginning “Figure 6” on line 328 by accident?

-The significant increase in phospho p65 in Fig Fig 5b is not reproduced in Fig 5d; do the authors have an explanation for this?

-The bands on Figure 5 aren’t labelled; which bands refer to what?

-I think the “In summary…” paragraph beginning line 349 is well written. However, I believe it would be better placed within the discussion.

-I don’t think the comment regarding a paper being from ‘Science’ (line 434), or the description of a paper as ‘outstanding’ is fitting. These sound like subjective perspectives which should be excluded.

Reviewer 2 Report

The authors assessed the inflammatory potential of the SARS-CoV-2 S1 protein in the two cell lines. The study could contribute the understanding of the severe COVID-19 cases and might possibly provide an explanation of Long COVID Syndrome. The methods were well documented while the results were thoroughly discussed with other studies. I have some comments so that the manuscript could improve further.

Major comments:

- lines 121-122 and 382: need to list out the sequence of S1 protein and let the readers to know about the VOC/lineage

- the S1 or S protein of SARS-CoV-2 was well studied in different research groups, the limitation of the current study or further studies should focus on different S1 or S proteins to assess the inflammatory capability, it is well known that the latest circulating strain i.e. omicron is less severe than the older strains such as alpha, beta, delta, etc. the authors can discuss this issue.

Minor comments:

- line 82: missing letter for S, should be S1

- line 98: the term GI was first mentioned here, need to spell it in full, alternatively, maybe it can be added in line 94, next to the word ‘gastrointestinal’

- line 102: the term BB barrier was first mentioned here, need to spell it in full

- line 106: the term BBB was first mentioned here, need to spell it in full

- line 385: typo ‘ACE2+’, the correct one should be A549+

Declaration:

The figures 2 and 3 could not be shown when I reviewed the manuscript, however, it will not affect my judgment of the manuscript.

Reviewer 3 Report

The manuscript by Forsyth and coauthors is well written and touches on a very acute topic at the moment - the pathway analysis of SARS-CoV-2 Spike protein-induced injury. 

1. The authors should justify, why they used A549+ lung epithelial cancer cells instead of regular human pulmonary alveolar epithelial cells.

2. Tha data shows that spike S1 activation of ERK 1/2 MAPK in alveolar epithelial cells. In the recent paper "Alcohol increases lung ACE2 expression and exacerbates SARS-CoV-2 Spike protein subunit 1-induced acute lung injury in K18-hACE2 transgenic mice" (Solopov et al, Am. J. Pathology, 2022) authors report that Spike protein S1 by did not activate ERK 1/2 in mice lungs.  Authors should explain that. 

Reviewer 4 Report

Results should be presented in a more concise way.

Some sentences in the introduction and in the discussion suggest that "long COVID" could be due to circulating protein spike. In my knowledge, nothing in the litterature supports this hypothesis.

Round 2

Reviewer 1 Report

Firstly, I now think the manuscript reads much better. The reader is led through the data in a clear manner and the conclusions are more restrained. Nevertheless, there are a few queries which I believe should be considered prior to acceptance for publication.

It may be as a result of so many track changes that it is hard to decipher, but is there a difference between A549 and A549 human lung Type 2 epithelial cells? I’m assuming these are standard A549 cells, in which case simply refer to them throughout as A549 (or A549+ for the overexpressing version). Alternatively, if there is a difference, then this should be described in the methods, and in the results/discussion as appropriate.

One of the interesting observations is that the inhibitor reduces the expression of certain cytokines. Whilst this is referred to, there is no discussion as to what might cause this. A brief insertion postulating a reason could be inserted around line 670.

The discussion can sometimes sound repetitive. For example, lines 764-765

“This suggests that targeting the MAPK pathway 764 could be a viable approach to blunt the cytokine storm induced by acute COVID-19 and 765 PASC.”

And lines 781-782

“MAPK inhibition could be a possible treatment target for 781 COVID-19 and PACS”.

The discussion would be tighter if such repetition was removed.

In my opinion, the paragraph of lines 813-827 would be better placed within the introduction.

Author Response

see attached PDF

Reviewer 4 Report

Major comments:

In this paper, the authors describe that IL-1 production in the supernatant of A549+ is increased after incubation with S1 protein, and that this increase is antagonized by the MEK1/2 inhibitor, PD98059. In contrast, they observed that IL-13 IL-8 and TNF alpha production was increased by the S1 protein, but they observed that the production of these cytokines is reduced by PD98059. Therefore, the production of IL-1 in the presence of PD98059 alone (basal production of IL-1 in the presence of PD98059) is mandatory to confirm that the decrease they observed is not due to an effect of PD 98059 on the basal production of IL-1, as observed for the other cytokine.

Second, they conclude thatthe activation of MEK1/2 is linked to the activation of NF-KappaB. This was observed in one series of experiments, but it was not confirmed in another series (figure 3d). Additional experiments are clearly required to conclude if there is really an activation of NF-KappaB or not.

It is mandatory to indicate the differences between the sequence of S1 protein used in this study and the VOC 

A more concise form should improve the paper: for exemple, line 66 to 68 ("corona appearance") in the introduction is useless. It is largely explained in the material and methods that you used the Meso scale to quantify cytokines, therefore it is useless to indicate in the results (line 249-250) that cytokines were quantified using Meso scale. In the beginning of the discussion, you summarized the results on MEK activation on A549+ celles (lines 393-395), that is repeated lines 467 -468 ...The discission should be more synthetic. 

Author Response

see attached PDF
